# Echocardiography-Assessed Changes of Left and Right Ventricular Cardiac Function May Correlate with Progression of Advanced Lung Cancer—A Generating Hypothesis Study

**DOI:** 10.3390/cancers14194770

**Published:** 2022-09-29

**Authors:** Sabina Mędrek, Sebastian Szmit

**Affiliations:** 1Department of Cardiology, Subcarpathian Oncological Center, 36-200 Brzozów, Poland; 2Department of Pulmonary Circulation, Thromboembolic Diseases and Cardiology, Centre of Postgraduate Medical Education, European Health Centre, 05-400 Otwock, Poland

**Keywords:** lung cancer, cancer disease progression, echocardiography, cardiac function, cardio-oncology

## Abstract

**Simple Summary:**

Lung cancer often coexists with cardiovascular diseases. Due to its specific location, lung cancer affects pulmonary circulation and leads to symptoms of dyspnea. If lung cancer is in an inoperable stage, the early response to cancer treatment determines prognosis. There are studies showing that advanced cancer, especially lung cancer, can be interpreted as advanced heart failure. This publication confirms that the progression of advanced inoperable lung cancer affects a number of parameters of right and left ventricular function. Importantly, the diagnosis of tumor progression may occur at the same time as the diagnosis of a cardiotoxic effect of cancer treatment. Cardio-oncology specialists should plan further research into these correlations and possible preventive strategies or care management.

**Abstract:**

Advanced lung cancer causes damage to lung tissue and the alveolar–capillary barrier, leading to changes in pulmonary circulation and cardiac function. This observational study included 75 patients with inoperable lung cancer. Two echocardiographic assessments were performed: one before the initiation of systemic anticancer therapy and another after the first radiological evaluation of the efficacy of anticancer treatment. In retrospective analysis, diagnosis of early cancer progression was associated significantly (*p* < 0.05) with some echocardiographic changes: a decrease in EF of at least 5 percentage points (OR = 5.78), an increase in LV GLS of 3 percentage points (OR = 3.81), an increase in E/E′ ratio of at least 3.25 (OR = 3.39), as well as a decrease in RV free wall GLS of at least 4 percentage points (OR = 4.9) and an increase in FAC of at least 4.1 percentage points (OR = 4.9). Cancer therapeutics-related cardiac dysfunction was diagnosed in accordance with the definition of the International Cardio-Oncology Society and was found more frequently in patients with radiologically confirmed lung cancer disease progression (*p* = 0.003). In further prospective studies, the hypothesis about the possible coexistence of the cardiotoxic effect of cancer therapy and cardiac dysfunction related to the progression of inoperable lung cancer should be clarified.

## 1. Introduction

Non-small cell lung cancer (NSCLC) is the most common lung cancer, accounting for 80–90% of cases, while the incidence of small cell lung cancer (SCLC) seems to be declining. Additionally, adenocarcinoma appears to be increasingly common as opposed to squamous cell carcinoma (SCC) [1]. Regardless of the histopathological diagnosis, lung cancer remains the main cause of oncological mortality and is diagnosed at an advanced stage ineligible for surgery in a high percentage of cases; therefore, the management is based on systemic treatment [2].

The search for prognostic and predictive factors is crucial in modern oncology, with key importance attributed to the identification of treatment resistance factors. In order to detect early progression or resistance to treatment, the first radiological follow-up is recommended after 2 or 3 cycles of chemotherapy or immunotherapy. It is also recommended to use the same imaging modality that was used for baseline staging. Computed tomography (CT) is the method of choice. Positron emission tomography (PET) is not routinely recommended due to its high sensitivity but relatively low specificity. The revised version of the Response Evaluation Criteria in Solid Tumours (RECIST 1.1) [3] is recommended. Despite such clear guidelines, there are many doubts about the accuracy of RECIST, e.g., in patients receiving EGFR/ALK-TKI therapy. Another problem is the phenomenon of pseudoprogression observed during immunotherapy. Therefore, alternative assessment methods, such as the immune-related RECIST (irRECIST) [4], immune-RECIST (iRECIST) [5], and immune-modified RECIST (imRECIST) [6], are being discussed. Early response to cancer therapy can determine survival in many cases of lung cancer.

Cardio-oncology problems have not been well known in lung cancer, especially at advances clinical stages. Given that lung cancer causes damage to lung tissue and the alveolar–capillary barrier, giving rise to secondary changes in the pulmonary circulation, it can be hypothesized that lung cancer progression may correlate with altered right and left ventricular function. Baseline echocardiographic parameters in patients with advanced lung cancer correlate with performance status and may predict overall survival time [7]. However, additional cardiotoxic effects of cancer treatment have not been well documented in lung cancer patients, especially in relations to the outcome of advanced disease [8]. The aim of this study was to determine a correlation between echocardiography-assessed changes of right and left cardiac function and radiological signs of progression of advanced inoperable lung cancer during first-line therapy.

## 2. Materials and Methods

During this prospective observational study in 75 patients with histopathologically confirmed inoperable lung cancer, two echocardiographic assessments were performed: one before the initiation of systemic anticancer therapy and another after the first CT scan assessing the efficacy of anticancer treatment.

The study included inoperable lung cancer disease either due to locally advanced stages or the presence of distant metastases. Initially, the clinical stage of lung cancer was evaluated by a multidisciplinary team including independent oncologists and radiologists. Anticancer treatment was carried out according to the guidelines of the Polish Society of Clinical Oncology [9]. Radiologists assessed the malignancy according to RECIST criteria.

This was a prospective pilot study based on diagnostic imaging performed at a single oncology center, with all echocardiographic examinations performed by the same experienced investigator, who had no influence on oncological decisions regarding patients, such as choice of anticancer therapy. The echocardiographer had no knowledge of the CT findings at the time of echocardiography examination. Another investigator provided independent factual oversight as to the quality of the examinations performed.

Echocardiographic diagnosis was performed according to the recommendations of the American Society of Echocardiography and the Section of Echocardiography of the Polish Cardiac Society [10,11].

The study received approval from the Bioethics Committee (No. 236/KBL/OIL/2018 dated 11 December 2018). The first and the last patient were included in the study on 23 January 2019 and 7 May 2020, respectively.

### Statistical Analysis

All nominal parameters were determined as frequencies; parameters on a continuous scale were determined by the arithmetic mean (in the case of normal distribution) or by the median and quartiles (in the case of distributions other than normal). Changes in individual parameters of left and right ventricular function were assessed, obtaining the difference (∆) between the values of the second versus the first echocardiographic examination. Logistic regression analysis was used to determine the relationship between the diagnosis of cancer progression on the first CT scan and echocardiographic or clinical parameters. The lower and upper quartile of the difference from each echocardiographic parameter were used as the points representing the differentiating criteria. A chi-squared test of independence was used to check for a correlation between diagnoses of cardiotoxicity and cancer progression.

## 3. Results

The majority of patients enrolled in the study, i.e., 60 patients (80%), were diagnosed with NSCLC compared to only 15 patients (20%) with SCLC. Clinical stage IV (metastatic disease) was diagnosed in 40 patients (53.33%). Such data reflect the worldwide situation of frequency of histopathological diagnoses in advanced lung cancer. Many patients had significant comorbidities, which is also typical in the population of patients with advanced lung cancer. Each patient had excluded any clinical features of infection such as pneumonia (confirmed additionally by computed tomography), sepsis, and exacerbation of obstructive pulmonary disease. Detailed characteristics of the patients are shown in Table 1.

Anticancer treatment included:Platinum-based doublet chemotherapy: 54 patients (72%);Cytotoxic monotherapy: 11 patients (14.67%);Pembrolizumab monotherapy: 7 patients (9.33%);Targeted therapy: 3 patients (4%).

Immunotherapy and targeted therapies were used according to the clinical predictive factors obtained (molecular testing for PDL1, EGFR, and ALK). In the case of chemotherapy, monotherapy was used in patients with worse performance status or significant comorbidities.

The first follow-up CT scan was performed after a median time of 96 days, and the first follow-up echocardiogram was performed after a median time of 98 days.

The first radiological assessment using computed tomography to evaluate the efficacy of anticancer treatment yielded the following results: complete response to treatment in 2 patients (2.67%), partial response to treatment in 35 patients (46.67%), stabilization of the disease in 22 patients (29.33%), and tumor progression according to RECIST in 16 patients (21.33%).

Accepting that the prognosis of patients with inoperable lung cancer is determined not only by histopathological, molecular diagnosis, and appropriately selected anticancer treatment—but also by the response to this treatment (especially early)—can determine survival, diagnosis of radiological progression was referred to for the main types of cancer therapy: 8 of 54 (14.8%) patients experienced lung cancer progression during treatment with platinum-based doublet chemotherapy, 7 of 11 (63.6%) patients during cytotoxic monotherapy, 1 of 7 (14.3%) patients on pembrolizumab monotherapy and no patients among those on targeted therapy.

Of the clinical factors, the occurrence of pleural effusion correlated with early cancer progression (Table 2). Deteriorated ECOG performance status, increased pain medication use, occurrence of anemia or neutropenia, mild pericardial effusion, and increased heart rate did not correlate with early cancer progression.

Changes in echocardiographic parameters of left and right ventricular function were calculated as the difference (∆) between the value at the time of follow-up CT and the baseline value. When upper and lower quartiles were used as criteria for changes, it was found that a decrease in EF of at least 5 percentage points (∆EF < −5, i.e., below the lower quartile), an increase in LV GLS of 3 percentage points (above the upper quartile), an increase in E/E′ ratio of at least 3.25 (above the upper quartile), a decrease in RV free wall GLS of at least 4 percentage points (below the lower quartile), and an increase in FAC of at least 4.1 percentage points (above the upper quartile) correlated with cancer progression at the first follow-up CT scan. Interestingly, none of the patients with increased EF (∆EF > 0) presented with cancer progression (Table 3).

The new criteria of cardiovascular toxicities of cancer therapies proposed most recently by the International Cardio-Oncology Society are based on EF and GLS [12]. They were used to identify a possible correlation with the observed signs of lung cancer disease progression (Table 4). We recognized 24 cases of CTRCD (cancer-therapeutics-related cardiac dysfunction): 4 as severe (final EF < 40%), 4 as moderate (decrease of EF by 10 percentage points to value 40–49% or any decrease of EF to 40–49% with deterioration of GLS > 15% from baseline), and 16 as mild (final EF ≥ 50% and deterioration of GLS > 15% from baseline). This was clearly reported in our previous study [7]. We revealed that the diagnosis of CTRCD correlated positively with radiological diagnosis of lung cancer disease progression.

## 4. Discussion

Lack of response to first-line treatment is a major problem in oncology. This may be indicative of the aggressiveness of the cancer itself. In metastatic/advanced NSCLC, this may be the lack of response to chemotherapy [13]. The problem of early progression during immunotherapy used as first- or second-line NSCLC treatment is even more thoroughly discussed [14] and may take the form of hyperprogression [15,16]. In the case of targeted therapies, primary resistance arising from various mutations may be also the clinical problem in planning effective sequence of cancer therapy [17]. The causes of resistance to systemic therapy are discussed in SCLC [18]. The earliest possible diagnosis of tumor progression enables appropriate qualification for the next line of treatment. The condition, however, is that the general performance status still allows for another line of treatment despite the observed tumor progression. Our study is the first to demonstrate echocardiographic signs of cancer progression. The question is whether such echocardiographic changes indicate only an increased size of the lung tumor, which affects the left and right ventricular function, or whether it is a more multifactorial process resulting from the progression of generalized malignant disease.

Echocardiographic assessment of left ventricular EF and GLS has been the cornerstone of the diagnosis of cardiotoxicity for a long time [19]. Experts suggest that cardiac monitoring during potentially cardiotoxic therapy should be based on echocardiography and biomarkers such as troponin and natriuretic peptides [20,21]. However, such formal monitoring recommendations for lung cancer have not been proposed to date. None of the cytostatics used in lung cancer are included in the list of drugs causing heart failure in the latest document of the European Society of Cardiology [22]. Although cardiovascular complications can be expected during systemic therapy in lung cancer, heart failure with reduced left ventricular ejection fraction is rare [23]. In this pilot study, we used only echocardiography. It would be difficult to monitor changes in cardiac biomarkers, which would probably already be elevated at baseline before anticancer treatment due to the frequent co-occurrence of various cardiovascular diseases, COPD and kidney disease. Our finding of significant changes in left ventricular EF and GLS indicates that the potential cardiotoxic effects of individual anticancer drugs should be taken into account in further research. However, in the light of our results, it is important to remember that changes in all echocardiographic parameters in advanced lung cancer may be caused by the progression of the tumor itself. Finally, it should be remembered that diagnoses of CTRCD and cancer progression may correlate in some lung cancer patients.

Thus, our study is another indication that advanced cancer should be regarded as the equivalent of advanced heart failure [24]. The echocardiographic changes we observed seem to confirm this. Lung cancer progression correlates with deteriorated LV function, which can be explained by the disturbed body’s homeostasis. It is possible that molecular and humoral mechanisms triggered by tumor progression induce cardiac damage. Impaired filling of the left ventricle is also observed, resulting in increased diastolic dysfunction (increased E/E′ ratio). There are also changes in the right ventricle—FAC increases, and the right ventricular free wall strain becomes more negative. It seems that these changes should be explained by the increased work of the right ventricle to overcome additional resistance in pulmonary circulation caused not only by the increasing tumor mass, but also by both inflammation and fibrosis within the surrounding pulmonary tissue. It seems important that changes in E/E′ ratio, FAC, RV free wall strain do not correlate with CTRCD and may help in differential diagnosis.

Our results should be interpreted with caution. This was only a pilot study in a very small group. However, we used very accurate diagnostic echocardiography. Further studies should be planned in larger populations. There should be dedicated studies for SCLC and NSCLC separately, and in fact, patients should receive uniform treatment, e.g., chemotherapy alone, immunotherapy alone, or specific targeted treatment alone. The clinical value of such work would be very high and could change the standards of monitoring these patients, which are currently based on CT. Echocardiography is a non-invasive method and causes no inconvenience for the patient; therefore, it could be performed much more often, and when parameters suggesting progression occur, CT could be performed. This would allow for an earlier intervention with ineffective systemic cancer treatment. Separate examinations should probably be performed for metastatic cancer and for locally advanced disease where radical treatment is still possible. Another element is the obligatory inclusion of cardiotoxicity biomarkers (troponin, natiuretic peptides), which have gained diagnostic and prognostic importance in cardio-oncology [25,26]. However, we should remember that cancer patients tend to have higher hemodynamic parameters, such as blood pressure (BP), cardiac output (CO), stroke volume (SV), and the maximal rate of pressure rise during isovolumic contraction (dP/dtmax), compared to healthy subjects and those with heart failure [27]. Thus, echocardiographic monitoring seems highly important to oncology. Our work may be a signal for cardio-oncologists to use echocardiography not only to monitor the cardiotoxic effects of anticancer drugs, but also to look for indicators of anticancer treatment inefficacy.

It should be highlighted that our study included advanced lung cancer patients with different important comorbidities, which were presented in Table 1. The control of status of such concomitant morbidities may impact prognosis and the results of echocardiographic parameters of right and left ventricular function. It is important that any clinical features of infection (such as pneumonia, sepsis, etc.) were excluded because firstly, they could also influence on echocardiographic cardiac results, and secondly, they are classical contraindications for the beginning of systemic cancer therapy. However, each case of cancer is a chronic inflammatory disease [28]. This is discussed in many cardio-oncology studies, especially including advanced cancer disease and the cardiac site effects of immune checkpoint inhibitors [29]. Even assuming that such an inflammatory process may affect the results of echocardiographic examinations, it does not change the diagnostic value of our study, because our pathophysiological hypothesis assumes that the ineffectiveness of systemic therapy, and thus tumor progression, have an impact on heart function [30]. Hence, in our opinion, cardiotoxicity of cancer therapy may correlate with the progression of cancer disease, especially in lung cancer. Cardiac biomarkers predict not only cardiovascular mortality but overall mortality [31]. Severe cardiotoxicity assessed using echocardiography was associated with higher risk of all-cause mortality [32]. Even changes in coagulation, such as decrease or increase in D-dimer concentration, can correlate with risk of progression of lung cancer and may be a prognostic factor for overall survival in lung cancer [33]. Patients experiencing cancer progression may be more sensitive to the cardiotoxic effects of chemotherapy, probably because their immune system and any protective barriers may be weaker [34]. On the other hand, there are single cases in which the effective treatment of cancer disease may improve the function of the heart [35]. We agree with proposed a new classification of cardio-oncology syndromes [36]. Based on our findings, the progressive development of cancer leads to cardiovascular disease, and this direct effect can be called type I cardio-oncology syndrome, as was suggested. Therefore, our assumption is that advanced neoplastic disease, such as lung cancer in particular, is a clinical condition that corresponds to advanced heart failure, and the clinical course, including prognosis, can be based on electrocardiografic assessment [37]. We suggest adding echocardiography for such cardio-oncology evaluation to prediction of the ineffectiveness of cancer therapy. In each advanced cancer disease, it will be difficult to make clear differential diagnosis with type II cardio-oncology syndrome, with cancer-associated treatments causing cardiovascular disease via indirect effects [36]. However, till today nobody has demonstrated the usefulness of echocardiography in monitoring of lung cancer patients. In the case of modern immunotherapy (like immune checkpoint inhibitors used in lung cancer), echocardiography turned out to be ineffective in monitoring cardiotoxicity [38]. It has not been clearly confirmed whether other drugs recommended in lung cancer can damage myocardial function. Further research and validation are required for our hypothesis.

This study is of a pilot nature. Cardio-oncology is a very difficult scientific field because in determining the final prognosis of patients, it must take into account two types of predictors: (1) oncological related to the cancer disease (histological diagnosis, molecular findings, clinical stage, etc.) and the effectiveness of cancer treatment (objective response, resistance to treatment, moment of progression: early vs. late, etc.) and also (2) cardiological factors associated with pre-existing cardiovascular disease as well as the effect of the neoplastic disease on the cardiovascular system (prothrombotic effect, direct or indirect influence on the heart function) and cancer treatment with potential cardiotoxic effect. Many modifiers of LV and RV function can be observed in oncology. The current study may define a new understanding of cancer-therapy-related cardiovascular disease and advise a new purpose for further prospective clinical trials.

The important limitation of the study was that echocardiography was used as only one cardiovascular technique for the evaluation of observed lung cancer patients. However, it is important to highlight the newest guidelines on cardio-oncology published by the European Society of Cardiology (ESC), which recommend echocardiography as the first-line modality for the assessment of cardiac function in patients with cancer [39]. The ESC guidelines treat biomarker evaluation as complementary to echocardiographic assessment. Cardiac biomarkers are obviously a component of the diagnosis of moderate or mild asymptomatic CTRCD. Data on the importance of biomarkers for cancer therapy related cardiovascular toxicity (CTR-CVT) risk stratification are limited before cancer therapy [40]. They can be a reference point for the subsequent increase induced by the toxicity of oncology drugs, but they can also identify patients with indications for further extended cardiological management. Baseline biomarkers may be associated with the probability of all-cause mortality, which was confirmed in patients with different cancer diseases [31]. The large registry CARDIOTOX in cancer patients receiving different oncological treatments showed no relationship between baseline biomarkers (NT-proBNP, troponin) and the development of severe CTRCD [32]. Perhaps, higher all-cause mortality should be explained mainly by earlier disease progression in patients with elevated biomarkers.

## 5. Conclusions

Changes in some parameters of left and right ventricular cardiac function may be a consequence of the progression of inoperable lung cancer during systemic therapy. It seems very important that CTRCD can coexist with the lack of cancer treatment efficacy, which would indicate that there is an overlapping effect related to the cardiac toxicity of lung cancer treatment and an effect related to the progression of advanced lung cancer disease.

## Figures and Tables

**Table 1 cancers-14-04770-t001:** Baseline characteristics of 75 patients with lung cancer evaluated by echocardiography.

Parameters	Characteristic by Numbers
Sex	
women	21 (28.0%)
men	54 (72.0%)
Age	Mean ± SD: 67.31 ± 7.94
BMI (kg/m^2^)	Mean ± SD: 25.46 ± 4.36
Never smokers	12 (16.0%)
Performance status (ECOG, Eastern Cooperative Oncology Group)	
0	23 (30.67%)
1	42 (56.0%)
2	10 (13.33%)
Comorbidities	
Arterial hypertension	40 (53.33%)
Chronic obstructive pulmonary disease	22 (29.33%)
Chronic coronary syndrome	15 (20.0%)
Diabetes mellitus	11 (14.67%)
Venous thromboembolic disease	8 (10.67%)
Hypothyrosis	7 (9.33%)
Chronic renal disease	2 (2.67%)

**Table 2 cancers-14-04770-t002:** Clinical possible predictors of lung cancer progression.

Possible Predictors	Univariable Analysis		
	OR	95% CI	*p*-Value
New pleural effusion	13.38	1.24–144.81	0.03
Deterioration of ECOG	0.33	0.04–2.87	0.30
New drug needed for pain control	0.56	0.11–2.88	0.48
New anemia	2.51	0.80–7.87	0.11
New neutropenia	0.76	0.23–2.54	0.66
New hemodynamic insignificant pericardial effusion	2.13	0.54–8.43	0.28
Increase in heart rate (HR)	1.42	0.46–4.41	0.53

**Table 3 cancers-14-04770-t003:** Diagnosis of early progression in first CT evaluation in relation to changes of echocardiography parameters. Univariate logistic regression.

Echocardiography Parameters		Lower and Upper Quartiles as Criteria	Univariable Analysis		
			OR	95% CI	*p*-Value
Left ventricular (LV) function	EF (%)	∆ < −5	5.78	1.60–20.90	0.007
		∆ > 0	-	-	-
	LV GLS (%)	∆ < −2	0.74	0.18–3.05	0.67
		∆ > 3	3.81	1.13–12.9	0.03
	E (cm/s)	∆ < −6.1	0.68	0.17–2.77	0.58
		∆ > 14.7	2.35	0.70–7.92	0.16
	E′ (cm/s)	∆ < −2.3	2.53	0.69–9.22	0.15
		∆ > 1.82	0.38	0.08–1.93	0.24
	E/E′ ratio	∆ < −1.91	0.68	0.17–2.77	0.58
		∆ > 3.25	3.39	1.02–11.33	0.04
Right ventricular (RV) function	TAPSE (mm)	∆ < −4	1.98	0.56–7.03	0.28
		∆ > 2	0.68	0.17–2.77	0.58
	RVSP (mmHg)	∆ < −3	1.18	0.32–4.37	0.8
		∆ > 13	1.18	0.32–4.37	0.8
	RV GLS (%)	∆ < −5	2.94	0.85–10.16	0.08
		∆ > 7	0.68	0.17–2.77	0.58
	RV free wall strain	∆ < −4	4.9	1.48–16.48	0.009
		∆> 7	0.68	0.17–2.77	0.58
	RV end-diastolic area (cm^2^)	∆ < −0.8	0.74	0.18–3.05	0.67
		∆ > 3.7	2.62	0.77–8.92	0.12
	RV FAC (%)	∆ < −2.2	1.61	0.46–5.58	0.45
		∆ > 4.1	4.9	1.46–16.48	0.009
	RV S′ (cm/s)	∆ < −2.2	1.61	0.46–5.58	0.45
		∆ > 0	1.01	0.25–4.06	0.99

Legends: EF—ejection fraction; LV GLS—left ventricle global longitudinal strain; E—early diastolic transmitral flow velocity; E′—early diastolic mitral annular velocity; TAPSE—tricuspid annular place systolic excursion, RVSP—right ventricular systolic pressure; RV GLS—right ventricle global longitudinal strain; FAC—fractional area change; S′—peak systolic tissue velocity at the tricuspid annulus; The delta symbol (∆) denotes the increment of any physical quantity, that is, the difference between the final and initial values.

**Table 4 cancers-14-04770-t004:** Correlation between diagnosis of CTRCD (cancer-therapeutics-related cardiac dysfunction) and different probable signs of lung cancer disease progression.

	CTRCD (*n* = 24)	No CTRCD (*n* = 51)	*p*-Value
Progression of lung cancer disease according to computed tomography (*n* = 16)	10 (41.67%)	6 (11.76%)	0.003
New pleural effusion (*n* = 4)	3 (12.5%)	1 (1.96%)	0.18
Deterioration of performance status (*n* = 11)	3 (12.5%)	8 (15.69%)	0.99
New drug needed for pain control (*n* = 14)	2 (8.33%)	12 (23.53%)	0.21
New anemia (*n* = 29)	8 (33.33%)	21 (41.18%)	0.52
New neutropenia (*n* = 27)	4 (16.67%)	23 (45.10%)	0.03
New hemodynamic insignificant pericardial effusion (*n* = 12)	6 (25%)	6 (11.76%)	0.14
Increase in heart rate (*n* = 37)	14 (58.33%)	23 (45.10%)	0.28
E/E′ ratio ∆ > 3.25 (*n* = 18)	8 (33.33%)	10 (19.61%)	0.19
RV free wall strain ∆ < −4 (*n* = 18)	5 (20.83%)	13 (25.49%)	0.88
RV FAC ∆ > 4.1 (*n* = 18)	7 (29.17%)	11 (21.57%)	0.47

Legends: CTRCD—cancer-therapeutics-related cardiac dysfunction; E—early diastolic transmitral flow velocity; E′—early diastolic mitral annular velocity; RV—right ventricle; FAC—fractional area change.

## Data Availability

Data may be available upon reasonable request.

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
