# Peer review of "Echocardiography-Assessed Changes of Left and Right Ventricular Cardiac Function May Correlate with Progression of Advanced Lung Cancer—A Generating Hypothesis Study"

_cancers, 2022, doi:10.3390/cancers14194770_

Round 1

Reviewer 1 Report

The topic of the paper is very important focused on a very interesting issue.

I should point out some concerns.

1. It's a pilot study with a small group of patients. It's known that there are many modifiers of LV and RV function of oncology group of aptients.

2. the table 1 should be modified. On the comorbidities part, the dislipidemia should be added.

3. The table 4 should also be modified. All abbreviations should be presented.

4. A limitation of the study was that echocardiography is the only used cardiovascular technique but it's important to point out that it's the only technique that used during follow up when it's necessary.

5. A prognostic analysis based on the chemotherapeutic agents should be added.

6. Additionally, the role of biomarkers should be discussed in this group of patients.

Author Response

Reviewer 1

The topic of the paper is very important focused on a very interesting issue.

I should point out some concerns.

  1. It's a pilot study with a small group of patients. It's known that there are many modifiers of LV and RV function of oncology group of patients.

We fully agree with the reviewer. We added the following paragraph in the discussion:

This study is of a pilot nature. Cardio-oncology is a very difficult scientific field because in determining the final prognosis of patients, it must take into account two types of the predictors: (1) oncological related to the cancer disease (histological diagnosis, molecular findings, clinical stage, etc.) and the effectiveness of cancer treatment (objective response, resistance to treatment, moment of progression: early vs late, etc.) and also (2) cardiological factors associated with pre-existing cardiovascular disease as well as the effect of the neoplastic disease on the cardiovascular system (prothrombotic effect, direct or indirect influence on the heart function) and cancer treatment with potential cardiotoxic effect. Many modifiers of LV and RV function can be observed in oncology. The current study may define a new understanding of cancer therapy related cardiovascular disease and advise a new purpose for further prospective clinical trials.  

  1. the table 1 should be modified. On the comorbidities part, the dislipidemia should be added.

Number of patients with dyslipidemia [ 22   (29.33%) ]  has been added to the Table 1.

  1. The table 4 should also be modified. All abbreviations should be presented.

The tables 3 and 4 have been modified and all abbreviations have been presented.

  1. A limitation of the study was that echocardiography is the only used cardiovascular technique but it's important to point out that it's the only technique that used during follow up when it's necessary.

We would like to thank for this important comment. We added the following statement:

The importnat limitation of the study was that echocardiography was used as only one cardiovascular technique for evaluation of observed lung cancer patients. However it's important to highlight the newest guidelines on cardio-oncology published by European Society of Cardiology (ESC) recommends echocardiography as the first-line modality for the assessment of cardiac function in patients with cancer [39].

  1. A prognostic analysis based on the chemotherapeutic agents should be added.

Accepting that the prognosis of patients with inoperable lung cancer is determined not only by histopathological and molecular diagnosis and appropriately selected anticancer treatment, but also the response to this treatment (especially early) can determine survival, diagnosis of radiological progression was referred to the main types of cancer therapy: 8 of 54 (14.8%) patients experienced lung cancer progression during treatment with platinum-based doublet chemotherapy, 7 of 11 (63.6%) patients during cytotoxic monotherapy, 1 of 7 (14.3%) patient on pembrolizumab monotherapy and nobody amongs patients on targeted therapy.

  1. Additionally, the role of biomarkers should be discussed in this group of patients.

We have added the discussion about biomarkers based on the newest ESC guidelines on cardio-oncology.

The ESC guidelines treat biomarker evaluation as complementary to echocardiographic assessment. Cardiac biomarkers are obviously a component of the diagnosis of moderate or mild asymptomatic CTRCD. Data on the importance of biomarkers for cancer therapy related cardiovascular toxicity (CTR-CVT) risk stratification are limited before cancer therapy [40]. They can be a reference point for the subsequent increase induced by the toxicity of oncology drugs, but they can also identify patients with indications for further extended cardiological management. Baseline biomarkers may be associated with probability of all-cause mortality, which was confirmed in patients with different cancer diseases [31]. The large registry CARDIOTOX in cancer patients receiving different oncological treatments showed no relationship between baseline biomarkers (NT-proBNP, troponin) and the development of severe CTRCD [32]. Maybe higher all-cause mortality should be explained mainly by earlier disease progression in patients with elevated biomarkers.

Reviewer 2 Report

I suggest the publication in This current form

Author Response

Reviewer 2

I suggest the publication in This current form

Thank you very much for the positive evaluation of our study.

Reviewer 3 Report

Work on the borderline of many specialties. For this reason, it is difficult to interpret due to the multitude of factors that can affect the change of the described parameters. Nevertheless, as a pilot study, it contains interesting suggestions as to the direction of further research on a broader material .

Author Response

Reviewer 3

Work on the borderline of many specialties. For this reason, it is difficult to interpret due to the multitude of factors that can affect the change of the described parameters. Nevertheless, as a pilot study, it contains interesting suggestions as to the direction of further research on a broader material .

We fully agree with the reviewer. The study is of a pilot nature, hence a very heterogeneous group of patients. The manuscript is a continuation of an earlier publication in which we looked for prognostic factors in lung cancer. Now we have found the relationship between echocardiographic changes and the progression of advanced lung cancer. In our opinion, the current study may define a new understanding of cancer therapy related cardiovascular disease. However, further prospective trials will be needed on larger groups of patients with a specific histopathological and molecular diagnosis treated with uniform anticancer regimens. Please find the new text added to the discussion (highlighted by blue font).

Round 2

Reviewer 1 Report

 The authors have answered all the reviewers recommendations